# Methods for Testing the Subjective Visual Vertical during the Chronic Phase of Menière’s Disease

**DOI:** 10.3390/diagnostics11020249

**Published:** 2021-02-05

**Authors:** Samira Ira Zabaneh, Linda Josephine Voss, Agnieszka J. Szczepek, Heidi Olze, Katharina Stölzel

**Affiliations:** 1Department of Otorhinolaryngology, Head and Neck Surgery, Charité-Universitätsmedizin Berlin, Corporate Member of Freie Universität, Berlin Humboldt Universität zu Berlin and Berlin Institute of Health, Campus Charité Mitte, Chariteplatz 1, 10117 Berlin, Germany; samira-ira.zabaneh@charite.de (S.I.Z.); agnes.szczepek@charite.de (A.J.S.); heidi.olze@charite.de (H.O.); 2Department of Audiology and Phoniatrics, Charité-Universitätsmedizin Berlin, Corporate Member of Freie Universität, Berlin Humboldt Universität zu Berlin and Berlin Institute of Health, Campus Charité Mitte, Chariteplatz 1, 10117 Berlin, Germany; linda-josephine.voss@charite.de

**Keywords:** inner ear diagnostics, subjective visual vertical, Menière’s disease, bucket test, C-SVV^®^ goggles, betahistine

## Abstract

The subjective visual vertical (SVV) evaluates the function of the utricle, which, in patients with Menière’s disease (MD), can be affected by endolymphatic hydrops. This study aimed to determine the SVV in MD patients during the chronic phase of illness compared to healthy participants. The second aim was to compare the SVV measurement tools: the analog bucket test, digital bucket test, and C-SVV© goggles. The SVV scores differed significantly between MD patients and the control group for the analog bucket test (*p* < 0.001) and the C-SVV^®^ goggles (*p* = 0.028), but no significance was shown when using the digital bucket test (*p* = 0.062). When comparing the analog bucket test and the C-SVV^®^ goggles applying the calculated threshold (1.125° in analog bucket test, 2.5° in C-SVV^®^ goggles), the bucket test showed higher accuracy (bucket test 73.84%, C-SVV^®^ goggles 69.23%). When examining the influence of betahistine on SVV scores, there were no statistically significant differences in both the analog bucket test and C-SVV© goggles. We conclude that SVV test can be used as an additional tool to evaluate utricle function during the chronic phase of MD and that the analog bucket test produces the most reliable results. The intake of betahistine does not influence the perception of SVV.

## 1. Introduction

The subjective visual vertical (SVV) reflects the deviation of individual perception of the Earth-vertical line. Healthy participants estimate the SVV within 2° of a real vertical line, depending on the testing laboratory or laboratory conditions [1]. The determined SVV angle allows a measure of utricular function [2,3].

The otolith-ocular reflex originates in the utricle. Signal processing leads through the vestibular nerve to the primary vestibular nuclear complex, which is described as an internal model of space and verticality, which is updated via bottom-up and top-down processes [4,5]. Several methods are used to measure utricular function: subjective haptic vertical (SHV), ocular vestibular-evoked myogenic potential (oVEMP), measurement of torsional eye movements [5]. Unilateral testing of SVV is performed by unilateral centrifugation with simultaneous measurement of SVV or oVEMP [6,7].

Various methods are used to measure the SVV. During the standard approach, the participant stands in front of a hemispheric dome and places a bar into SVV [8]. This examination is stationary. Another method is the bucket test, developed more recently by Zwergal [9]. During that test, the participant looks into a plastic bucket with a dark diametric line marked at the bottom. The examiner rotates the bucket until the participant indicates that the SVV of this line is reached. A perpendicular line drawn on the outside of the bucket serves to evaluate the angle (degrees). For a digital measurement of the SVV, the C-SVV^®^ goggles (Diatec Diagnostic GmbH, Dortmund, Germany) are used. The SVV detects acute unilateral utricle lesions in a very rigid manner. Typically, central compensation leads to the normalization of the SVV [10].

SVV is used in the diagnostic panel for MD, vestibular neuritis, benign positional paroxysmal vertigo, or labyrinthitis [1] and was suggested for patients with multiple sclerosis [11] or orthostatic hypotension [12]. In a previous study, we evaluated the deviation of SVV in MD patients in correlation to the affected side. There was a 100% correlation between the side of the deviation and the affected ear in MD patients, indicating that an ipsilateral deviation of SVV in MD patients can be expected [13].

The typical symptoms of MD are vertigo, hearing loss, and tinnitus. In 2015, the Barany Society proposed new diagnostic criteria for MD that differentiate between *definite MD* and *probable MD.* In addition to two or more episodes of vertigo and fluctuating aural symptoms such as tinnitus or ear fullness, the diagnostic criteria of *definite MD* include low- to middle-tone unilateral hearing loss during or after the vertigo attack documented by pure tone audiometry. Hearing loss in the affected ear should be no less than 30 dB in at least two frequencies below 2000 Hz, compared to the contralateral ear. If both ears are affected, then the hearing loss should be at least 35 dB, affecting at least two frequencies below 2000 Hz [14]. Endolymphatic hydrops (E.H.) is considered to be a pathologic correlate of MD [15], and the utricle can be affected by this pathology. Kumagami et al. examined the SVV in MD patients during the attack. In 63.3% of cases, a tilt of SVV was detected, indicating utricular dysfunction in MD patients [16].

Betahistine is commonly used in Europe for MD treatment, although the evidence supporting its effectiveness is weak [17]. The proposed effect of betahistine is an antagonism of the H3-receptor and agonism of the H1-receptor, increasing cochlear blood flow, especially in the stria vascularis [18]. There are many positive reports, while there are also studies questioning the efficacy of betahistine. For example, the so-called BEMED trial found no statistically significant differences between betahistine and placebo [19]. At present, histidine is the only drug approved for MD therapy and, therefore, is of high clinical relevance.

This study aimed to evaluate the SVV as a further diagnostic parameter for Menière’s disease during the chronic phase of the condition. An additional aim was to identify the optimal tool for SVV examination. Usually, the auditory and vestibular properties of the inner ear are tested using audiometry, caloric testing, and video head impulse test (HIT). As complementary diagnostic tools, cervical vestibular evoked myogenic potentials (cVEMPs) for the saccule, and oVEMPs for the utricle are commonly used. Here, other examination methods of the utricle during the chronic phase of MD, not requiring specialized equipment, were tested. The SVV measurement was performed using two types of buckets (analog and digital) and the C-SVV^®^ goggles. MD patients (in the chronic phase) and control subjects were analyzed. The primary research question was whether the SVV differs between MD patients and the control subjects. The secondary goal was to determine whether using the analog and digital buckets and the C-SVV^®^-goggles yields comparable results. Lastly, the influence of betahistine intake on the SVV results was examined.

## 2. Materials and Methods

### 2.1. Study Design

The local ethics committee approved this prospective study (E.A. 1/276/15; 3 December 2015). In addition to 34 patients with MD (followed-up for 2 years) enrolled in primary and secondary health providers, 40 healthy subjects consisting of volunteers were recruited as a control group. The inclusion criteria for the control group were age between 20 and 80 years and no neuro-otologic diseases. The patients were diagnosed with at least “probable MD” (24 of the 26 participants included had definite Menière’s disease), according to the classification of the Barany Society:Two or more episodes of vertigo or dizziness, each lasting for 20 min to 24 h;Fluctuating aural symptoms (hearing loss, tinnitus, or fullness) in the reported ear;Not better accounted for by another vestibular diagnosis;Patients were not examined during an acute phase of MD. Other exclusion criteria in both groups were pregnancy, lactation, and diseases of the cervical spine.

### 2.2. SVV Bucket Test

The SVV bucket test was used in two variations. The set-up of the examination with the analog and digital bucket is shown in Figure 1a–e. The digital bucket test used the Visual Vertical app. During the test, the bucket was fixed on a tripod to obtain reproducible results.

During the analog bucket test, the participants were sitting in front of the bucket looking into it to avoid environmental influence. The participant’s face had to be placed precisely at the top of the bucket. First, the examiner tilted the bucket by 45° to the right and then slowly tilted it back until the participant signaled “stop” when the subjective visual vertical was reached. The participant could make small changes. The same procedure was then used on the left side.

For the digital bucket test, we used the Visual Vertical app, with the smartphone fixed on the bottom of the inside of the bucket. The app generated a bright red line (10.4 × 0.3 cm). The bucket was tilted 45° to the examiner’s right, and then the participants had 20 s to tilt the bucket to the line that seemed vertical to them. The examiner could read the number of degrees off the vertical after the time had expired. The same procedure was used on the left side.

### 2.3. C-SVV^®^ Goggles

Using the C-SVV^®^ goggles, the SVV could be determined digitally. The participant put on the goggles, which completely darkened the visual field. Then, a tilted orange line was shown to the participants. The participants could move this line to the perceived vertical by remote control. The goggles were connected to a computer using the program “OtoAccess^TM^” (Figure 1f,g), enabling recording the SVV values.

### 2.4. Statistical Analysis

Statistical analyses were performed with IBM SPSS Statistics, version 25 (IBM Corp., Armonk, NY, USA). First, the variables were tested for a normal distribution using descriptive statistics and the Kolmogorov–Smirnov test. When the significance of this test or the skewness >1 indicated a non-normal distribution, then nonparametric tests were used for further analysis. Here, the Mann–Whitney *U*-test was used for independent samples. In the case of a normal distribution, the *t*-test was used.

Linear regression analysis was used to adjust the results for age and sex. Significance was set at a *p*-value ≤ 0.05. When analyzing subgroups, the Bonferroni correction was applied.

## 3. Results

### 3.1. Participants and Eligibility Criteria

The recruited MD group consisted of 34 patients, whereas the control group included 40 participants. After extended diagnostics, eight MD patients and one control subject were excluded as not meeting the inclusion criteria, leaving 26 MD patients and 39 participants in the control group. There was a statistically significant difference in age between the groups (*p* < 0.001), with the control subjects being younger than the MD patients (Table 1). No significant difference in sex distribution was seen (*p* = 0.100).

### 3.2. Comparison of Patients with MD and of the Control Group Concerning the SVV Measurement Using the Bucket

The SVV scores obtained from the analog bucket test differed significantly between MD patients and the control group (Figure 2) (*p* < 0.001, Mann–Whitney *U*-test). The median SVV score of the MD group was 1.125°, while that of the control group was 0.75°. That difference remained statistically significant after adjusting for age and sex. The digital bucket test using the Visual Vertical app indicated no significant differences between the MD patients and the control group (Figure 2) (*p* = 0.062).

### 3.3. Comparison of MD Patients and the Control Group Regarding SVV Measurement Using the C-SVV^®^ Goggles

There was a significant difference in SVV scores between MD patients and the control group (Figure 2) (*p* = 0.028). Linear regression analysis could not rule out the influence of age difference between the groups. This aspect is covered in more detail in the discussion.

### 3.4. Comparison of Sensitivity and Specificity of Bucket Test and C-SVV^®^

For a direct comparison between the bucket test and digital C-SVV^®^, we used the ROC (receiver operating characteristic) curves (Figure 3). The greater the area under the curve (AUC), the better the accuracy of the test. We demonstrated superior accuracy of the bucket test, with greater AUC than that of C-SVV^®^.

### 3.5. Between-Group Comparison of Bucket Test Results and C-SVV^®^ Goggles Results

In each study, a new threshold can be derived from the AUC by determining the point with the largest area. We determined that the threshold value for the bucket test was 1.125°, meaning that an SVV greater than 1.125° can be considered pathologic. The new threshold value for the digital C-SVV^®^ was 2.5°. A confusion matrix identified a threshold accuracy of 73.84% for the bucket test and 69.23% for the C-SVV^®^ goggles (Table 2a,b).

### 3.6. SVV Deviation and Betahistine Intake by MD Patients

A secondary endpoint of the study was comparing MD patients taking betahistine (*n* = 16) versus MD patients not taking the drug (*n* = 10). The Mann–Whitney *U*-test determined no statistically significant differences in SVV deviation between patient groups, as per the analog bucket test (*p* = 0.182) or the C-SVV^®^ goggles (Figure 4).

## 4. Discussion

This work’s primary research aim was to determine whether the SVV differs between MD patients and the control subjects. The obtained results indicate a statistically significant difference in SVV results between MD patients and the control subjects. In addition, significant differences in outcome between using analog and digital buckets and the C-SVV^®^-goggles were determined, suggesting an advantage of using analog bucket test. Lastly, it was determined that the betahistine intake has no critical influence on the SVV results.

In this study, the SVV scores were assessed as a further diagnostic criterion for MD. The SVV scores were evaluated in 26 MD patients during the chronic phase of the disease, of which 24 were diagnosed with definite MD. In addition, 39 healthy individuals (control group) were subjected to SVV testing.

Previously published research indicated that 63.3% of MD patients [13] had a pathologic deviation in SVV when tested during the acute phase. In most of the patients, the SVV values returned to the baseline after a couple of weeks. Furthermore, no patient had a pathological SVV before the MD attack, indicating that during the chronic phase of MD, the SVV scores remain unchanged [16]. In our sample, the median time from the last MD attack was 33.5 days. However, in contrast to Kumagami et al., the SVV test indicated significant differences in SVV. This discrepancy between the studies’ outcomes can be attributed to a different SVV measurement technique, as Kumagami used a joystick with a digital set-up in a darkened examination room, whereas we used a physical bucket and a natural light.

In their study, Baier et al. showed that brainstem lesions lead to SVV deviation and that the side of the deviation is dependent on the localization of the lesion. Lesions at the pontomesencephalic level lead to contraversive roll-tilt, whereas lesions at the pontomedullary regions are associated with ipsiversive roll-tilt of the SVV [20].

Corroborating our results, other studies examining the SVV in MD patients in the chronic phase (e.g., Pinar et al. [21]) determined the SVV of patients with peripheral (including MD patients) and central vestibulopathy found a significant difference between MD patients and healthy controls, with 25–50% of the patients with chronic dizziness having a pathological SVV or SVH (subjective visual horizontal). The examination was performed up to 2 months after an attack. The authors concluded that the otolith function should also be evaluated in the chronic phase of the disease [21].

The clinical guidelines of the American Academy of Otolaryngology and Head and Neck Surgery consider the vestibular examinations not necessary for the diagnosis of MD. That is justified by the fact that, due to the fluctuation of MD, no valid results can be expected. Moreover, the evidence from existing studies on vestibular diagnostics in MD is insufficient [22]. Additionally, the costs for both diagnostics and therapy can be very high, according to Basura et al. [23]. In our opinion, the determination of SVV with the analog bucket test as an additional diagnostic tool can be justified by the presented results. The method is rapid and inexpensive, particularly if the suspicion of MD could not be supported by clinical and audiometric means. Our results suggest that SVV can also be determined during the chronic phase of MD and that the bucket test and the C-SVV^®^ goggles are suitable for measurements of SVV in MD patients and control subjects.

By using the SVV, we aimed to evaluate the function of the utricle in MD patients. However, studies showed the saccule seems to be more involved in MD than utricle, according to a visualization of the hydrops by magnetic resonance imaging (MRI) [24,25,26]. In addition to the SVV, Kahn et al. have demonstrated that combining MRI, VEMPs, and vHIT can be even more helpful in evaluating vestibular function in MD patients [26].

The comparison between the bucket test with and without the Visual Vertical application indicated that, as opposed to the analog bucket test, the digital bucket test yielded no statistically significant differences between MD and healthy controls. The reason for this discrepancy could be a lack of standard app values, which could not be identified in the latest literature. It was assumed that there might be a measurement inaccuracy with the app since other parameters (e.g., set-up of the bucket, the examiner, and the distance between the participant and bucket) remained the same as during the analog bucket test. Additionally, another reason might be that the analog bucket test was more comfortable than using the digital bucket with the Visual Vertical app for the participants. In the analog bucket test, one could concentrate on the black line. In the digital bucket test with the app, there was a red line on a gray background plus the smartphone and an acoustic signal counting every second, which could have been disturbing to the person being tested.

Although the digital C-SVV^®^ goggles could distinguish sufficiently between MD patients and healthy controls, the analog bucket test had a higher sensitivity and specificity in the detection of abnormal findings. The new threshold value in this study derived from the ROC curve was 1.125° for the analog bucket test. The accuracy (73.84%), specificity (89.7%), and sensitivity (50%) were superior to the C-SVV^®^ goggles, and thus, we consider the bucket test to be eligible for distinguishing between individuals with and without MD.

The SVV test performed using the C-SVV^®^ goggles also showed a significant difference between MD patients and healthy controls; however, the median angle obtained using the C-SVV^®^ goggles was 2.09°, which was above the threshold of 2°. Upon applying a new threshold of 2.5°, we achieved a sensitivity of 46.2%, meaning that half of the MD patients during the chronic phase have average test results.

Our present results differ from those presented by Michelson et al. [27], who, using exclusively healthy participants, compared the SVV bucket test with the C-SVV^®^ goggles. The average SVV value was 0.81° for the bucket test and −0.29° for the C-SVV goggles. The authors concluded that C-SVV^®^ was a more precise and reliable tool for measuring the SVV than the bucket test because of the higher absolute test–retest stability. On the other hand, they only compared the two tests, and only the healthy subjects were examined. Therefore, a direct comparison with our study might be difficult since we also had patients with MD. In addition, the mean of the subjective verticals in healthy volunteers varies from laboratory to laboratory, and there are various standard values.

Zwergal et al., who first described the bucket test, obtained intratest reliability of 0.92 after 10 experiments. The authors compared the tests using the hemispheric dome method. There was no statistically significant difference between the examination tools. Consequently, the authors considered that the SVV testing with the bucket was fast and reliable while being easily accessible and inexpensive [9].

Our ultimate aim was to determine the influence of betahistine intake on the perception of SVV. In our sample, there were no differences between patients taking and not taking betahistine. A possible explanation might be that the H3- and H1-receptors, which are a target of betahistine, are located mainly in the saccule and not the utricle. In addition, the results of the BEMED trial study showed no statistically significant difference between betahistine and placebo [19]. Another explanation is that the H3/H1 receptors might play a crucial role during endolymphatic hydrops corresponding with MD attacks, but not during the sustained periods of the disease.

### Study Limitations

Our study is not free of limitations. The first drawback of the study is the small sample size, which is a consequence of low MD prevalence (between 3.5 and 513/100,000 persons) [28]. With such a low prevalence, the collection of a higher number of cases could be achieved over a much more extended period. The second limitation is a significant age difference between MD patients and the control group. This difference remained statistically relevant for the C-SVV^®^ goggles after applying the linear regression analysis, so it cannot be ruled out that age influences the SVV perception in a significant way. On the other hand, Sun et al. examined the SVV in a study population with a mean age of 77 years, and they concluded that the optimum diagnostic threshold should be 2°. We, therefore, assume that the SVV does not depend on age [29].

## 5. Conclusions

Presented research suggests that the SVV can be used as a rapid and inexpensive additional tool for the functional evaluation of the utricle during the chronic phase of MD. The analog bucket test was superior to the C-SVV^®^ goggles in our study. Moreover, the Visual Vertical app for the digital bucket test was not suitable for distinguishing between patients with MD and healthy participants. Finally, no significant influence of betahistine intake on SVV perception could be demonstrated.

## Figures and Tables

**Figure 1 diagnostics-11-00249-f001:**
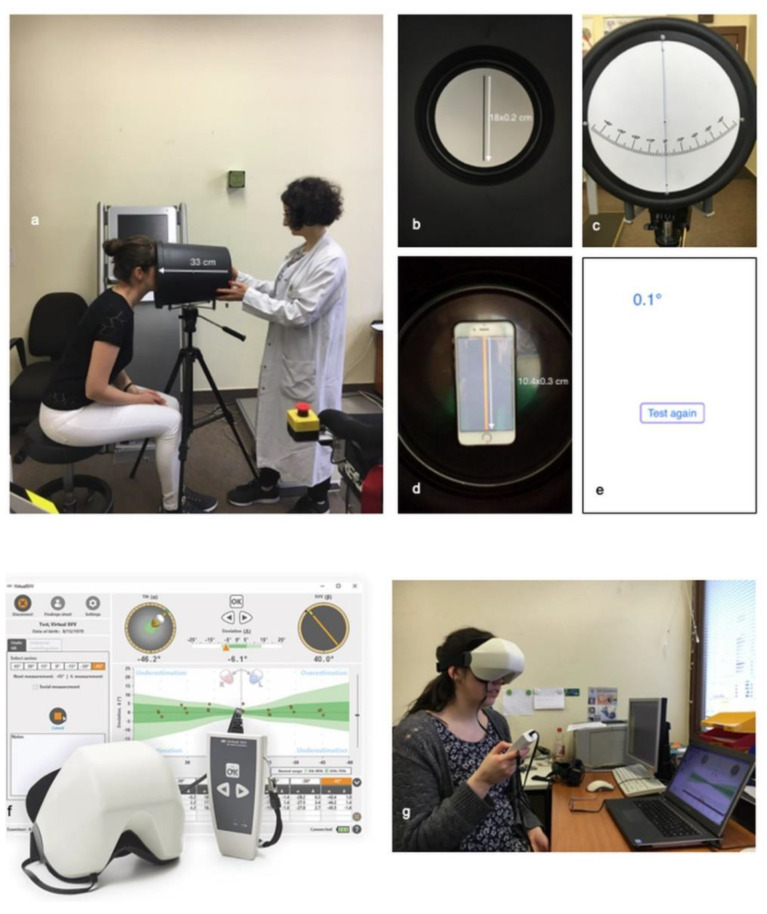
Examination set-up with the analog and digital bucket. (**a**) Examination of the subjective visual vertical (SVV) with the bucket test (distance from participant’s eyes to line on the bottom of the bucket 33 cm); (**b**) view of the analog bucket test for a participant (measures of the line: 18 × 0.2 cm); (**c**) view of the analog bucket test for an examiner; (**d**) view of the Visual Vertical app for a participant (measures of the line: 10.4 × 0.3 cm); (**e**) result of the Visual Vertical app as seen by the examiner; (**f**) C-SVV^®^ goggles (image reproduced with permission from Diatec Diagnostics GmbH and was reused from [13] with permission of Springer); (**g**) examination of SVV with the C-SVV^®^ goggles.

**Figure 2 diagnostics-11-00249-f002:**
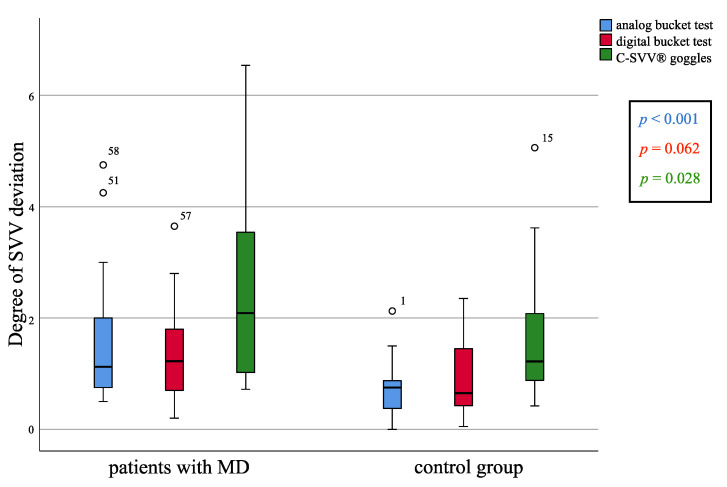
Comparison of the subjective visual vertical (SVV) scores between Meniere’s disease (MD) patients and the control group measured with the analog bucket test, the digital bucket test using the Visual Vertical app, and the C-SVV^®^ goggles. The difference was statistically significant for the analog bucket test (*p* < 0.001, Mann–Whitney *U*-test) and the C-SVV^®^ goggles (*p* = 0.028, Mann–Whitney *U*-test), but not statistically significant for the digital bucket test (*p* = 0.062, Mann–Whitney *U*-test).

**Figure 3 diagnostics-11-00249-f003:**
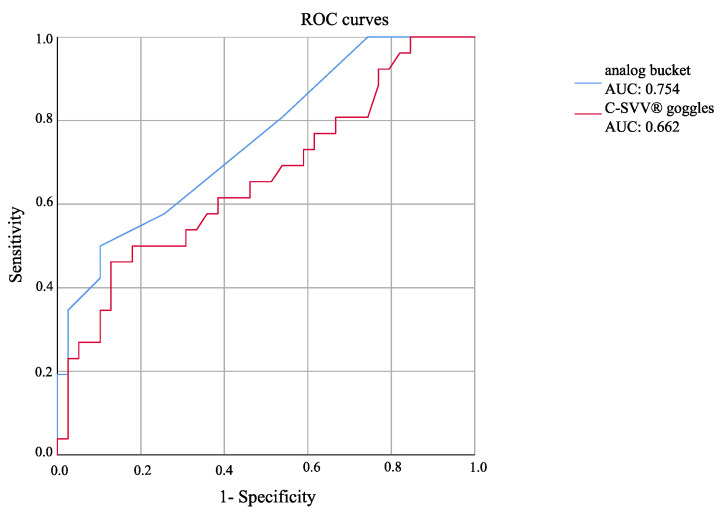
ROC curves for the analog bucket test and C-SVV^®^ goggles.

**Figure 4 diagnostics-11-00249-f004:**
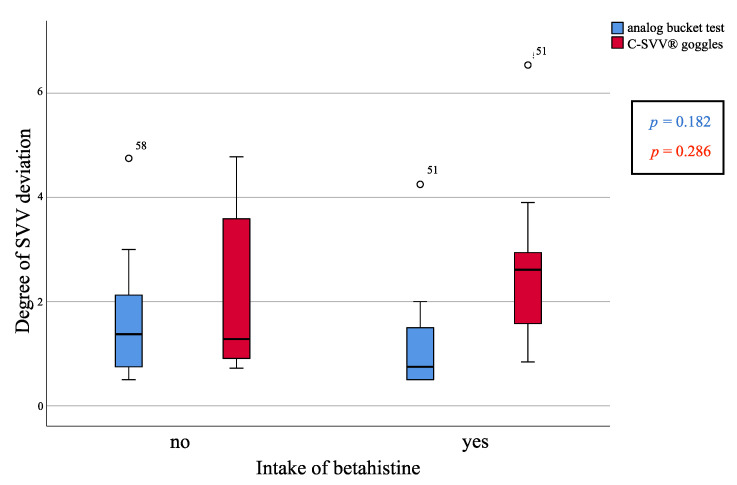
Differences in the subjective visual vertical (SVV) deviation within Meniere’s disease patients dependent on betahistine intake measured with the analog bucket test and C-SVV^®^ goggles. The differences were not statistically significant (*p* = 0.182 and *p* = 0.286, respectively, Mann–Whitney *U*-test).

**Table 1 diagnostics-11-00249-t001:** Descriptive statistics for MD patients and controls.

**Parameters Measured**	**Patients with MD (*n* = 26)**	**Control Group (*n* = 39)**
Age, median (range), years	54.50 (25–75)	25.00 (20–59)
MD classification (Barany), definite/probable	24/2	-
Sex, female/male	15/11	30/9
Average DHI	39.28	-
**Analog Bucket Test, Degrees**		
Average (±standard deviation)	1.500 (±1.134)	0.686 (±0.446)
Median	1.125	0.750
Difference in averages	–0.814	
**Digital Bucket Test (Visual Vertical App), Degrees**		
Average (±standard deviation)	1.346 (±0.836)	0.968 (0.701)
Median	1.225	0.650
Difference in averages	–0.378	
**C-SVV^®^ Goggles, Degrees**		
Average (±standard deviation)	2.391 (±1.567)	1.550 (±0.986)
Median	2.090	1.220
Difference in averages	–0.841	
**AUC of ROC curves**		
Bucket test analog	0.754	
C-SVV^®^-goggles	0.662	
**Influence of Intake of Betahistine (Analog Bucket Test), Degrees**		
**No**		
Average (±standard deviation)	1.656 (±1.125)	-
Median	1.375	-
**Yes**		
Average (±standard deviation)	1.250 (±1.161)	-
Median	0.750	-
**Influence of Intake of Betahistine (C-SVV^®^ Goggles), Degrees**		
**No**		
Average (±standard deviation)	2.191 (±1.559)	-
Median	1.280	-
**Yes**		
Average (±standard deviation)	2.710 (±1.607)	-
Median	2.610	-

The average and median values for every test performed are shown for each group. AUC: area under the curve; DHI, Dizziness Handicap Inventory MD: Menière’s disease; ROC: receiver operating characteristic.

**Table 2 diagnostics-11-00249-t002:** Tables of confusion for the analog bucket and C-SVV^®^ goggles tests.

(a) Confusion matrix for the analog bucket test with the new pathologic threshold of >1.125°.			
**Predicted Condition**	**Patients with Meniere’s Disease**	**Control Group**	**Accuracy**
Negative	T.N.: 13 (50%)	T.N.: 35 (89.7%)	NPV = 72.9%
Positive	F.P.: 13 (50%)	F.P.: 4 (10.3%)	PPV = 76.5%
	Specificity: 50%	Specificity: 89.7%	Accuracy = **73.84%**
(**b**) Confusion matrix for the C-SVV^®^ goggles with the new pathologic threshold of >2.5°.			
**Predicted Condition**	**Patients with Meniere’s Disease**	**Control Group**	**Accuracy**
Negative	F.N.: 14 (53.8%)	T.N.: 34 (87.2%)	NPV =70.2%
Positive	T.P.: 12 (46.2%)	F.P.: 5 (12.8%)	PPV = 70.6%
	Sensitivity: 46.2%	Specificity: 87.2%	Accuracy = **69.23%**

T.N.: true negative; F.N.: false negative; NPV: negative predictive value; F.P.: false positive; T.P.: true positive; PPV: positive predictive value.

## Data Availability

Primary data are available upon request.

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
