# Peer review of "Methods for Testing the Subjective Visual Vertical during the Chronic Phase of Menière’s Disease"

_diagnostics, 2021, doi:10.3390/diagnostics11020249_

Round 1

Reviewer 1 Report

This paper describes the updated subjective visual vertical test using goggle. Actually, saccule is more involved than utricle in Meniere's disease. 

Please additional description in discussion for the readers.

Correlation of utricualr function test using MRI delayed inner ear images, OVEMP, and vHIT. I think that combined test will be better than single SVV using goggle. 

Author Response

Reply to Reviewer 1

Thank you very much for all your comments. We addressed the points raised in the Discussion.

This paper describes the updated subjective visual vertical test using goggle. Actually, saccule is more involved than utricle in Meniere's disease. 

To address this point, three papers showing that saccule is more involved than utricle in Meniere’s disease were added to our manuscript:

  1. Sugimoto S, Yoshida T, Teranishi M, Kobayashi M, Shimono M, Naganawa S, et al. Significance of Endolymphatic Hydrops Herniation Into the Semicircular Canals Detected on MRI. Otol Neurotol. 2018;39(10):1229-34.

  1. Attye A, Eliezer M, Medici M, Tropres I, Dumas G, Krainik A, et al. In vivo imaging of saccular hydrops in humans reflects sensorineural hearing loss rather than Meniere's disease symptoms. Eur Radiol. 2018;28(7):2916-22.

  1. Kahn L, Hautefort C, Guichard JP, Toupet M, Jourdaine C, Vitaux H, et al. Relationship between video head impulse test, ocular and cervical vestibular evoked myogenic potentials, and compartmental magnetic resonance imaging classification in meniere's disease. Laryngoscope. 2020;130(7):E444-E52

Please additional description in discussion for the readers.

Additional description was added using added references:

„By using the SVV, we aimed to evaluate the function of the utricle in MD patients. However, studies showed the saccule seems to be more involved in MD than utricle, ac-cording to a visualization of the hydrops by MRI (22-24).”

(Line 252-254)

Correlation of utricular function test using MRI delayed inner ear images, OVEMP, and vHIT. I think that combined test will be better than single SVV using goggle. 

The above suggestion was addressed in the discussion:

“In addition to the SVV, Kahn et al. have demonstrated that combining MRI, VEMPs, and vHIT can be even more helpful in evaluating vestibular function in MD patients (24)”

(Line 254-256)

Reviewer 2 Report

The authors present another manuscript on the topic of Mb. Meniere chronic phase and measurement of SVV (by various methods). This paper is a complement to previous results published in 2019 in the HNO.

The authors thoroughly explained that this is not a similarity between the manuscript from 2019 and the current paper.

The study is handled very precisely. 

For daily medical practice, it is important to know that a bucket examination is sufficient.

My question to the authors: please add information on which other diagnoses (other than Mb. Menier) is the importance of the SVV examination for practice?

My Recommendation: Accept after minor revision

Author Response

Reply to Reviewer 2

The authors appreciate the constructive criticism of Reviewer 2.

My question to the authors: please add information on which other diagnoses (other than Mb. Menier) is the importance of the SVV examination for practice?

We added the following information about the importance of the SVV examination in clinical practice:

“SVV is used in the diagnostic panel for MD, vestibular neuritis, benign positional paroxysmal vertigo, or labyrinthitis (1) and was suggested for patients with multiple sclerosis (11) or orthostatic hypotension (12).”

(Line 53-55)

And:

„In their study, Baier et al. showed that brainstem lesions lead to SVV deviation and that the side of the deviation is dependent on the localization of the lesion. Lesions at the pontomesencephalic level lead to contraversive roll-tilt, whereas lesions at the pontome-dullary regions are associated with ipsiversive roll-tilt of the SVV (18).”

(Line 230-233)

  1. Chetana N, Jayesh R. Subjective Visual Vertical in Various Vestibular Disorders by Using a Simple Bucket Test. Indian J Otolaryngol Head Neck Surg. 2015;67(2):180-4.
  2. Klatt BN, Sparto PJ, Terhorst L, Winser S, Heyman R, Whitney SL. Relationship between subjective visual vertical and balance in individuals with multiple sclerosis. Physiotherapy research international : the journal for researchers and clinicians in physical therapy. 2019;24(1):e1757.
  3. Aoki M. The impaired subjective perception of verticality independent of peripheral vestibular function in dizzy elderly with orthostatic hypotension. Aging clinical and experimental research. 2017;29(4):647-53.
  4. Baier B, Thömke F, Wilting J, Heinze C, Geber C, Dieterich M. A pathway in the brainstem for roll-tilt of the subjective visual vertical: evidence from a lesion-behavior mapping study. J Neurosci. 2012;32(43):14854-8.